# COARSENING TO CONCEAL: ENABLING PRIVACY-PRESERVING FEDERATED LEARNING FOR GRAPH DATA

## ABSTRACT

With the escalating demand for privacy-preserving machine learning, federated learning (FL) stands out by enabling collaboration among decentralized entities. Utilizing graph representations of data enhances learning for graph-level tasks, crucial for FL with data distributed across local repositories. Despite its benefits, stringent privacy regulations often compromise FL's performance. Previous methods aimed at ensuring privacy introduce performance degradation and computational overhead. In response to these challenges, we propose using graph coarsening—a simple yet effective method—to enhance the security and privacy of FL on graph data. Our approach posits that graph coarsening alone can suffice for privacy guarantees, as model parameters obtained from training on the coarsened graph effectively conceal sensitive information susceptible to privacy attacks. Through comprehensive application and analysis, we demonstrate the efficacy of graph coarsening within an FL setup, taking both the graph matrix and node features as input, and jointly learning the coarsened graph matrix and feature matrix while ensuring desired properties. The resultant coarsened graph representations are then utilized to train model parameters, subsequently communicated within an FL framework for downstream tasks such as classification. Extensive experimentation across various datasets confirms that graph coarsening ensures privacy while enhancing performance with minimal trade-offs compared to traditional differential privacy (DP) methods without adding extra complexity overhead.

## 1 INTRODUCTION

Federated learning (FL) is a distributed machine learning approach enabling multiple decentralized entities to collaboratively train a shared model without exchanging their local data, thus preserving data privacy by keeping raw data localized and only sharing model updates [41]. This paradigm offers enhanced data privacy, reduced latency, and the ability to leverage diverse datasets from multiple sources, resulting in robust and generalizable models [43]. However, FL faces significant privacy challenges, as the exchange of model updates can still inadvertently leak sensitive information through inference attacks [26]. Addressing these concerns, methods like differential privacy (DP) [34; 13], secure multi-party computation [28; 16], and homomorphic encryption [24; 15] have been proposed, each with its trade-offs [19] such as reduced model performance, increased computational complexity, and added communication overhead [37; 33]. Graph representation is crucial in capturing relationships between entities, aiding in tasks like graph classification and prediction [40]. For instance, in molecular research [9], graphs predict properties or classify enzymes [12], while in social networks [22], graphs facilitate community detection. In the context of graph data, FL involves clients each with a local graph represented by a set of nodes and edges [11]. Each client collaboratively trains a global graph neural network (GNN) model by sharing model parameters or gradients rather than raw graph data [3].

Privacy attacks in FL threaten the confidentiality of graph data, even when raw data isn't exchanged. Adversaries can infer sensitive information from shared model updates, such as through gradient inversion attacks where gradients are used to reconstruct private data [31]. For example, in a federated graph machine learning scenario where healthcare institutions collaborate to predict patient outcomes, an attacker could infer sensitive patient information from the model updates [30; 17; 14]. This highlights the urgent need for robust privacy-preserving techniques in FL. Current methods for privacy preservation in FL are categorized into data manipulation and model gradient manipulation

techniques [32; 30]. Data manipulation includes secure multi-party computation (SMPC) and homomorphic encryption (HE), which ensure computations are carried out without revealing raw data [7]. Model gradient manipulation, particularly DP, adds controlled noise to model updates, with differential privacy stochastic gradient descent (DP-SGD) being a notable method [35; 27]. However, these methods face significant limitations: the trade-off between privacy and model accuracy increased computational complexity, and potential performance degradation [1]. For instance, in fraud detection using graph data, the noise added by DP can impair the detection accuracy, revealing the limitations of DP in complex FL scenarios [25]. Addressing these challenges is crucial to advancing privacy-preserving FL for graph data.

Data-based approaches for privacy preservation in FL, including data condensation and data reduction, have been effective for non-graph and graph data [10]. Data condensation (DC) involves summarizing a large dataset into a smaller synthetic dataset that retains essential statistical properties, providing privacy by limiting the impact of individual samples on model parameters, effectively offering DP [8]. Data reduction techniques reduce the dimensionality or amount of data, thereby minimizing the exposure of sensitive information [5]. However, these methods face limitations when applied to graph data due to its intricate structure. To address this, we propose using graph coarsening as a privacy measure. Graph coarsening simplifies a graph by merging nodes and edges, creating a smaller version that preserves essential structural properties [4]. Reconstructing the original graph from its coarsened version is challenging, thus protecting sensitive information. In our approach, clients coarsen their local graphs before training local models, which are then shared with the server for aggregation, maintaining privacy with minimal performance trade-offs. We focus on featured graph coarsening (FGC) [18] due to its simplicity and flexibility. FGC ensures that coarsened graphs retain necessary information, allowing for effective model training while ensuring privacy. This method integrates seamlessly into the FL framework, providing a robust solution for privacy preservation in federated graph machine learning.

Our main contributions are summarized below:

- We introduce Graph Coarsening for Privacy-Preserving Federated Learning (CPFL), a novel framework integrating graph coarsening techniques into FL to enhance privacy while maintaining performance balance.
- CPFL ensures privacy without additional communication overhead by training local models on coarsened graphs, minimizing data transmission between clients and the server.
- We built the connection between graph coarsening and differential privacy and validated our approach's robust privacy preservation and effectiveness across various scenarios, including cross-domain datasets in multi-client and multi-dataset settings.

## 2 BACKGROUND

### 2.1 FEDERATED GRAPH NEURAL NETWORK

In FL for graph data, GNNs are used for graph classification in a distributed environment where privacy and regulatory restrictions prevent the centralization of data [38]. This scenario involves either partitioning a single graph dataset or distributing multiple graph datasets across several edge servers or clients. Despite the inability to centralize data, collaborative training on this distributed data can yield more powerful and versatile graph models. Our work focuses on leveraging GNNs as the primary model for this collaborative effort in private and secure manner, applying them across various domains characterized by heterogeneous graph data [11].

We consider a scenario with $K$ clients, each possessing its own dataset $\mathcal{D}^{(k)} = (\mathcal{G}^{(k)}, \mathcal{Y}^{(k)})$. Here, $\mathcal{G}^{(k)} = (\mathcal{V}^{(k)}, \mathcal{E}^{(k)})$ represents the graph(s) within the dataset, with vertex and edge feature sets $X^{(k)} = \{x_m^{(k)}\}_{m \in \mathcal{V}^{(k)}}$ and $Z^{(k)} = \{e_{m,n}^{(k)}\}_{m,n \in \mathcal{V}^{(k)}}$. The corresponding label set is denoted as $\mathbf{Y}^{(k)}$. Each client trains a local GNN model on its data to learn graph representations and make predictions. To enhance their models, these clients collaborate via a central server, sharing their locally trained model parameters without disclosing their private data.

Here, GNNs are integrated into the FL paradigm. A typical GNN involves message propagation and neighbourhood aggregation, where each node iteratively collects information from its neighbours and combines it with its own to update its representation. This process for an $L$-layer GNN is expressed as:

$$h_v^{(l+1)} = \sigma \left( h_v^{(l)}, \text{agg} \left( \{ h_u^{(l)}; u \in \mathcal{N}_v \} \right) \right), \quad \forall l \in [L], \tag{1}$$

where $h_v^{(l)}$ is the representation of node $v$ at the $l$-th layer, and $h_v^{(0)} = x_v$ represents the initial node feature. $\mathcal{N}_v$ denotes the neighbors of node $v$, $\text{agg}(\cdot)$ is the aggregation function which varies with different GNN architectures, and $\sigma$ is an activation function.

The graph-level representation $h_G$ can be obtained by aggregating the node representations:

$$h_G = \text{readout}(\{ h_v; v \in \mathcal{V} \}), \tag{2}$$

where $\text{readout}(\cdot)$ can be implemented through methods like mean pooling or sum pooling, which aggregate the node embeddings into a single vector suitable for tasks such as graph classification.

To used GNNs in FL setting, the aggregation step involves combining the model updates from multiple clients into a single global model update. This aggregation can be performed using various strategies, including simple averaging, weighted averaging, adaptive methods like Adam, or other custom approaches. The general aggregation step is represented as:

$$\theta^{(t+1)} = \text{Aggregate} \left( \left\{ \theta_k^{(t)}, \mathcal{D}_k \right\}_{k=1}^{K} \right). \tag{3}$$

where $\theta^{(t+1)}$ is the aggregated global model parameter at round $t + 1$, $\{\theta_k^{(t)}\}_{k=1}^{K}$ represents the model parameter from the $K$ clients at round $t$, and $\mathcal{D}_k$ is the local dataset of client $k$. The function $\text{Aggregate}(\cdot)$ serves as a placeholder for the specific aggregation method used.

## 2.2 GRAPH COMPRESSION: PRIVACY-PRESERVING APPROACHES IN FL

In FL, multiple client devices collaboratively train a shared model while keeping their local data decentralized, enhancing data privacy by preventing direct data sharing. However, privacy risks persist as model updates can leak sensitive information through gradient inversion attacks [36]. To mitigate these risks, various data manipulation techniques, such as random sparsification and dataset condensation, have been proposed [23; 42]. Random sparsification involves randomly removing edges to achieve anonymity and can ensure DP while approximating the original graph's spectrum [6]. However, there is a trade-off between the degree of sparsification and the preservation of critical information, which impacts the graph's utility. Excessive edge removal can compromise the graph's structural integrity and informative value. Dataset condensation, on the other hand, transforms the dataset into a smaller, abstract representation while preserving essential features [8; 39]. In FL, condensed local graphs on client devices reduce the risk of sensitive information leakage during aggregation. By synthesizing representative data points instead of sharing actual data, condensation maintains privacy and mitigates re-identification risks.

Despite the advantages, these techniques have limitations. Sparsification selects existing graph elements, which may reduce interpretability and relevance to the original graph. Condensation, while effective, can struggle to balance the trade-off between privacy and the preservation of key information, leading to potential losses in data utility. Additionally, condensation methods can introduce artifacts that deviate from the original data's natural structure, impacting downstream machine learning tasks [10].

Graph coarsening presents a compelling alternative to sparsification and condensation for privacy preservation in FL. Graph coarsening aggregates graph elements into supernodes and superedges, making it challenging to reconstruct the original graph, thus enhancing privacy. The coarsened graph retains the critical information necessary for effective model training while significantly reducing the risk of sensitive information leakage. Moreover, graph coarsening can mitigate the issue of relating back to the original graph, which is a common concern with sparsification and condensation methods.

One established method for graph coarsening is FGC [18]. Given an original graph $\mathcal{G} = (\mathcal{V}, \mathcal{E}, X, L)$, where $X \in \mathbb{R}^{p \times n}$ denotes the feature matrix of $p$ nodes and $L \in \mathbb{R}^{p \times p}$ is the graph Laplacian, FGC aims to learn a coarsened graph $\tilde{\mathcal{G}} = (\tilde{\mathcal{V}}, \tilde{\mathcal{E}}, \tilde{X}, \tilde{\mathcal{L}})$ with $m$ supernodes. The optimization problem

FGC solves is formulated as follows:

$$\min_{\tilde{X},C} -\gamma \log \det(C^T L C + J) + \text{tr}(\tilde{X}^T C^T L C \tilde{X}) + \frac{\lambda}{2}\|C^T\|_{1,2}^2, \tag{4}$$

$$\text{s.t. } C \in \mathcal{S}_C = \left\{ C \geq 0 \mid \|[C^T]_i\|_2^2 \leq 1, \forall i = 1, \ldots, p \right\}, \quad X = C\tilde{X}, \tag{5}$$

where $C \in \mathbb{R}^{p \times m}$ is the coarsening matrix that maps the original graph to the coarsened graph, and $\tilde{X} \in \mathbb{R}^{m \times n}$ represents the feature matrix of the coarsened graph. The term $-\gamma \log \det(C^T \Theta C + J)$ ensures the connectivity of the coarsened graph, $\text{tr}(\tilde{X}^T C^T \Theta C \tilde{X})$ enforces smoothness in the feature mapping, and $\frac{\lambda}{2}\|C^T\|_{1,2}^2$ imposes desirable properties on the mapping matrix $C$. The constraint $X = C\tilde{X}$ denotes the feature mapping from the original graph to the coarsened graph.

The FGC formulation is a multiblock non-convex optimization problem, efficiently solved using a block successive upper bound minimization technique that iteratively updates variables in blocks, ensuring convergence while balancing graph structure preservation and size reduction. We chose FGC for our framework due to its unique strengths and its ability to guarantee similarity between the coarsened and original graphs. This is the first work to integrate graph coarsening with FL to ensure privacy, effectively handling heterogeneous datasets across different domains and distributions. While our CPFL pipeline can utilize any coarsening algorithm, FGC's model-agnostic, flexible, efficient, and simple nature makes it ideal for diverse FL applications, maintaining robust privacy and utility.

# 3 THE GRAPH COARSENING FOR PRIVACY-PRESERVING FEDERATED LEARNING (CPFL) FRAMEWORK

## 3.1 PRIVACY AND TRADE-OFFS IN GRAPH DATA REDUCTION TECHNIQUES

In FL, preserving data privacy while maintaining the utility of graph data for downstream tasks is a significant challenge. DP [27] provides a mathematical framework to ensure that the inclusion or exclusion of a single element (such as a node or edge in a graph) does not significantly affect the outcome of an analysis. Formally, a randomized algorithm $M$ satisfies $(\epsilon, \delta)$-DP if, for any two neighboring graphs $G$ and $G'$ (differing by one node or edge), and for any subset $S$ of the output space, the following inequality holds:

$$P[M(G) \in S] \leq e^\epsilon P[M(G') \in S] + \delta \tag{6}$$

Here, $\epsilon$ represents the privacy budget, quantifying the allowed privacy loss, and $\delta$ accounts for the probability of the privacy guarantee being broken.

However, traditional reduction techniques like sparsification and condensation, while enhancing privacy, often compromise the utility of the graph data. In tasks such as protein-protein interaction networks, road networks, and social network analysis, removing nodes or edges can strip away critical information, degrading the graph's utility. Condensation methods synthesize fake elements, potentially distorting the graph's original structure and affecting tasks that rely on precise interactions, such as drug discovery.

One promising solution here is graph coarsening which retains essential structural properties and minimises the risk of reconstructing the original graph. It preserves data utility while enhancing privacy, making it ideal for FL environments where both are critical. Visualization of graph coarsening is given in Figure 1.

## 3.2 PROBLEM FORMULATION

Organizations often leverage sensitive graph data to advance their research and services. For instance, pharmaceutical companies may use biological interaction networks to discover new drugs, and research institutions may analyze genetic data for disease prediction. However, using raw graph data ($\mathcal{G}$) for model training poses significant privacy risks, including susceptibility to membership inference attacks and potential data leaks during transmission to cloud servers, particularly from honest-but-curious operators. To mitigate these risks, a more secure protocol involves first transforming the data by generating a coarsened graph dataset ($\tilde{\mathcal{G}}$) from the original graph data ($\mathcal{G}$), which is then used for model training in downstream applications. The threat model can be formalized as follows:

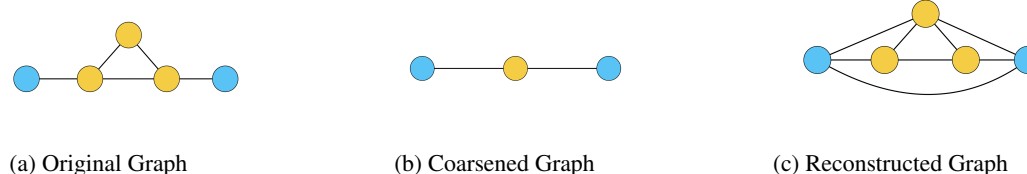

(a) Original Graph             (b) Coarsened Graph             (c) Reconstructed Graph

Figure 1: Visualization of Graph Coarsening: Illustrating the transformation from the original graph to a coarsened version, highlighting node merges and edge reductions.

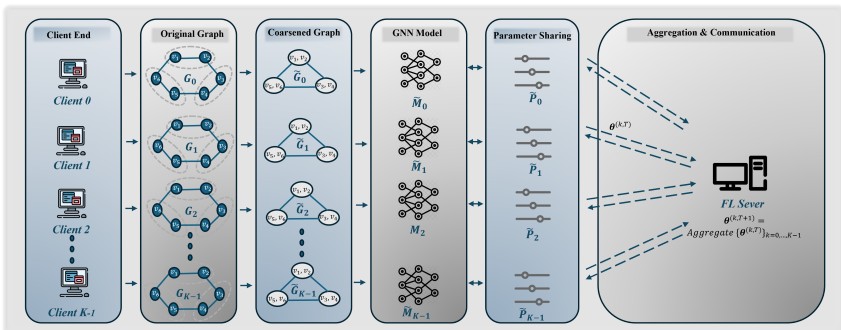

Figure 2: The CPFL Framework.

**Attacker's Objective** The attacker's goal is to determine if a specific node $v$ or edge $e$ is part of the original graph dataset $\mathcal{G}$.

**Attacker's Knowledge** We assume a robust malicious entity, such as an honest-but-curious server, who lacks direct access to $\mathcal{G}$ but has white-box access to both the coarsened graph dataset $\tilde{\mathcal{G}}$ derived from $\mathcal{G}$ and the model $f_{\tilde{\mathcal{G}}}$ trained on $\tilde{\mathcal{G}}$.

**Attacker's Capacity** The attacker possesses unlimited computational resources to generate shadow coarsened graph datasets from data with the same distribution as $\mathcal{G}$ and to train shadow models on them. It is important to note that white-box access to model parameters does not significantly aid MIA, thus we exclude other advantages conferred by white-box access to $f_{\tilde{\mathcal{G}}}$.

## 3.3   TECHNICAL DESIGN

We employ an FL framework where multiple clients with graph data collaborate via a central server, ensuring data remains decentralized and private. GNNs are ideal for this FL framework as their parameters encapsulate structural and feature information of graphs. The framework of CPFL is shown in Figure 2, where each client's graph data $\mathcal{G}_i = (\mathcal{V}_i, \mathcal{E}_i, X_i, Y_i)$ is coarsened to $\tilde{\mathcal{G}}_i = (\tilde{\mathcal{V}}_i, \tilde{\mathcal{E}}_i, \tilde{X}_i, \tilde{Y}_i)$ using a coarsening ratio $r$. This reduces data complexity while retaining essential structure and obscuring sensitive information.

Each client trains a local GNN model on $\tilde{\mathcal{D}}_k$ which is a set of $\tilde{\mathcal{G}}_i$ graphs each client has, resulting in model parameters $\theta_k$. At each communication round $t$, clients send their parameters $\theta_k^{(t)}$ to the server, which aggregates them to update the global model parameters $\theta^{(t+1)}$:

$$\theta^{(t+1)} = \text{Aggregate}\left(\left\{\theta_k^{(t)}, \tilde{\mathcal{D}}_k\right\}_{k=1}^{K}\right). \tag{7}$$

The updated global model is sent back to clients for further training. This process repeats until convergence. Aggregation can use strategies like weighted averaging, ensuring the global model reflects all clients' knowledge while maintaining privacy. Our method shows that graph coarsening in FL enhances privacy and maintains performance in graph-level tasks, achieving effective model

performance with minimal privacy risks and computational overhead compared to traditional methods like DP.

## 3.4 CONNECTION TO DP

Nodes and edges in a graph can contain highly sensitive information, such as social contacts, personal opinions, and private communication records. Node-Differential Privacy (Node-DP) and Edge-Differential Privacy (Edge-DP) offer rigorous theoretical guarantees to protect the privacy of these connections by limiting the influence of any single node or edge on the output [27]. This provides meaningful privacy protection in various applications.

In the context of graph coarsening, given a graph $\mathcal{G} = (V, E)$ algorithm aims to learn a coarsened graph $\tilde{\mathcal{G}}(\tilde{\mathcal{V}}, \tilde{\mathcal{E}})$ using a mapping matrix $C$. It is important to note that this reconstructed graph $\mathcal{G}_r(\mathcal{V}_r, \mathcal{E}_r)$ from $\tilde{\mathcal{G}}(\tilde{\mathcal{V}}, \tilde{\mathcal{E}})$ is just edges-added version of $\mathcal{G}(\mathcal{V}, \mathcal{E})$, having no discernible meaning to it. Based on the above-mentioned threat model, we will evaluate different attack scenarios to assess if graph coarsening is able to preserve privacy or not.

**Case 1**: If the attacker has access to model parameters of $k^{th}$ client $\theta_k^{(t)}$, it can only infer about the coarsened graph $\tilde{\mathcal{G}}$. Since the information about the nodes and edges between them is still not known, the privacy is preserved.

**Case 2**: If the attacker has access to model parameters of $k^{th}$ client $\theta_k^{(t)}$ and the loading matrix $C$ (which means coarsening ratio ($r$) is known), attacker can reconstruct the coarsened graph $\tilde{\mathcal{G}}$ and using this, it is possible to construct a $\mathcal{G}_r$ with same nodes as the original graph. However, the connection is still concealed thereby preserving edge-level privacy.

**Definition:** An algorithm $A$ satisfies $\varepsilon$-edge differential privacy ($\varepsilon$-edge DP), where $\varepsilon > 0$, if and only if for any two edge neighboring graphs $G$ and $\tilde{G}$ is satisfied

$$\forall T \subseteq \mathrm{Range}(A) : \Pr[A(\mathcal{G}) \in T] \le e^{\varepsilon} \Pr[A(\mathcal{G}_r) \in T] \tag{8}$$

where $\mathrm{Range}(A)$ denotes the set of all possible outputs of $A$.

**Discussion:** After coarsening, the graph can be reconstructed using the relation

$$L_r = P^T \tilde{L} P = P^T C^T L C P = (CP)^T L C P = (CC^{\dagger})^T L C C^{\dagger} \neq L \tag{9}$$

Since $CC^{\dagger} \neq I$ and it is a block diagonal matrix. This signifies that the reconstructed graph retains a similar number of nodes as the original graph, but contains more edges compared to the original graph. Thus we can say that the two dataset $\mathcal{G}(V, E)$ and $\mathcal{G}_r(V_r, E_r)$ are differentially private. Therefore, since we trained our model using the coarsened graph without knowledge of the coarsening ratio, it is impossible to revert to the original graph, ensuring that our dataset remains 100% private.

**Definition(Global $L_2$-sensitivity $\Delta_2$):** Let $f$ represent the aggregation function and $D$ (or $D'$) be the users' data. Let $X$ be the set of all neighbouring databases. We can define the (global) $L_2$-sensitivity of $f$ as:

$$\Delta_2(f) := \max_{D, D' \in X, D \simeq D'} \|f(D) - f(D')\|_2 \tag{10}$$

We note that the maximum is taken over all neighbouring pairs of datasets in $X$.

**Theorem:** Motivated by the $L_2$ sensitivity definition, in our case, we consider $D = L$, i.e., the original graph, and $D' = L_r = (CC^{\dagger})^T L C C^{\dagger}$ is reconstructed from the coarsened graph $\tilde{L}$. The $L_2$ sensitivity for our case is defined as follows:

$$\Delta_2 = \max_{D, D' \in X, D \simeq D'} \|L - (CC^{\dagger})^T L C C^{\dagger}\|_2 \tag{11}$$

The sensitivity $\Delta_2$ depends on the coarsening ratio $r = \frac{m}{p}$, where $p$ is the number of nodes in the original graph and $m$ is the number of nodes in the coarsened graph. Aggressive coarsening ($r < 0.5$) conceals almost 100% of the nodes, ensuring near-complete privacy and maximizing sensitivity. Decreasing $r$ increases $\Delta_2$. So, coarsening ratio ($r$) controls the level of privacy preserved as illustrated in later section.

**L$_2$ Sensitivity w.r.t. Features:** Consider $D = X \in \mathbb{R}^{P \times d}$ represent the feature matrix of the original graph and $D' = X_r \in \mathbb{R}^{P \times d}$ is the reconstructed features obtained from the coarsened features $\tilde{X} \in \mathbb{R}^{k \times d}$. The $L_2$ sensitivity with respect to feature is defined as:

$$\Delta_2 = \max_{D, D' \in X, D \simeq D'} \|X - X_r\|_F \tag{12}$$

$$\|X - X_r\|_F = \|X - C\tilde{X}\|_F = \|X - C(C^+X)\|_F = \|(I - CC^+)X\|_F \tag{13}$$

Note that $I - CC^+$ acts as a projection matrix that determines the reconstruction error introduced by coarsening. We can further bound this error using the properties of the Frobenius norm:

$$\|(I - CC^+)X\|_F \leq \|I - CC^+\|_F \|X\|_F \tag{14}$$

**Analysis Based on Coarsening Ratio:** Let $r = \frac{k}{P}$ denote the **coarsening ratio**, which measures the reduction in the number of nodes. Consider the following scenarios:

- $r \to 1$**:** When $r \to 1$, $k \approx P$, meaning the number of supernodes is almost equal to the number of original nodes. In this scenario, $C^+ \approx C^{-1}$, and hence $CC^+ \approx I$. This implies:

$$\|I - CC^+\|_F \to 0$$

  Thus, the reconstruction error $\|X - X_r\|_F$ is minimized, indicating that the coarsened graph retains almost all the information of the original graph.

- $r \to 0$**:** When $r \to 0$, $k \ll P$, meaning a large number of original nodes are mapped to a small set of supernodes. This results in a significant loss of information, making $CC^+$ far from being an identity matrix. In this case:

$$\|I - CC^+\|_F \gg 1$$

  This leads to a high reconstruction error, making the coarsened graph more sensitive to changes in the original graph structure.

**Sensitivity and Differential Privacy Implications:** As $r$ decreases, the reconstruction error increases, which in turn increases the sensitivity of the model output. This behavior can be linked to **differential privacy**, where sensitivity quantifies the impact of changes in the input on the output. High sensitivity requires adding more noise to achieve a given privacy budget $\epsilon$. Therefore, as the coarsening ratio decreases (i.e., as $r \to 0$), the sensitivity increases, making it harder to ensure strong privacy guarantees without significant utility loss.

In contrast, when $r$ is high, the sensitivity is low, which means less noise is needed to achieve the same privacy guarantee. This suggests that, depending on the coarsening ratio, the balance between privacy and utility can be controlled, making graph coarsening a potentially effective privacy-preserving mechanism in federated learning.

## 4 EXPERIMENTS

### 4.1 EXPERIMENTAL SETTINGS

**Datasets** We use 13 graph classification datasets from three domains: molecules (5), proteins (3), and social networks (5) [38]. Node features and graph labels vary across datasets, as detailed in Appendix A and B. We experiment with different coarsening ratios ($r$) and noise levels ($\epsilon$) to evaluate privacy using graph coarsening and DP-SGD. Performance is assessed by convergence improvements or degradations round-wise.

**Baselines** We first test self-training to see if decentralized learning improves with collaborative training, using FedAvg [21], FedProx [20], GCFL, and GCFL+ [38]. Each client trains a locally downloaded model without communication. The graph classification model used is a GIN, with architecture and hyper-parameters fixed across all baselines.

**Parameters** We use three-layer GINs with a hidden size of 64, batch size of 128, and Adam optimizer (learning rate 0.001, weight decay 5e-4). For FedProx, $\mu$ is 0.01, and the local epoch $E$ is 1. The coarsening ratio ($r$) and noise level ($\epsilon$) are varied across all settings. Experiments ran on 24GB NVIDIA TITAN RTX GPUs.

## 4.2 CLASSIFICATION ACCURACY: CCPL VS DP-SDG

We present classification accuracy using graph coarsening and differentially private stochastic gradient descent (DP-SGD) [27] in FL settings. DP-SGD, the gold standard for privacy-preserving machine learning, adds calibrated noise to gradients to protect individual data points, quantified by the privacy budget ($\epsilon$)—lower values indicate stronger privacy. We evaluated performance based on varying coarsening ratios ($r$) and privacy budgets ($\epsilon$), also assessing effectiveness in heterogeneous data settings by combining datasets from Molecules, Proteins, and Social Networks.

Our evaluation shows that graph coarsening often outperforms DP-SGD in maintaining classification accuracy across different datasets and settings. For example, in single-data multi-client settings (given in Table 1) like the PROTEINS dataset, graph coarsening achieved 0.62 accuracy with FedProx, compared to 0.62 with DP-SGD and 0.75 with the original dataset. In the IMDB-(B) dataset, graph coarsening achieved 0.72 with FedAvg, compared to 0.47 with DP-SGD. In the multi-data multi-client setting given in Table 2, for the Molecules group, although DP values indicated that FedProx handled privacy better with a DP of 0.67, the consistent performance of graph coarsening techniques underscores their viability in heterogeneous settings. In multi-data multi-client multi-domain settings shown in Table 3, graph coarsening showed significant advantages, such as in the MIX 2 setting (Molecules + Social Networks), where FedProx achieved 0.65 accuracy with a coarsening ratio of 0.1, compared to 0.55 with DP-SGD ($\epsilon = 8$).

While DP-SGD offers strong privacy protection, it often reduces classification performance. Graph coarsening, however, provides a balanced approach, preserving essential structural properties while offering privacy benefits with minimal impact on classification accuracy. This makes graph coarsening a compelling choice for privacy-preserving FL, especially where high model performance is critical.

**Preserving Properties of $\tilde{\mathcal{G}}$ in CPFL** Incorporating hyperbolic error (HE)[2] metrics into our FL analysis highlights the balance between utility and privacy in graph coarsening. Our results (given in Figure 3) show that GCFL and GCFL+ outperform FedAvg and FedProx, supported by low HE values indicating minimal distortion. For example, in the Proteins dataset, HE values for coarsening ratios of 0.3, 0.5, and 0.7 are 21.41, 18.18, and 14.69, respectively, demonstrating retained structural properties. Similar trends are observed in Molecules and Social Networks datasets. These low HE values confirm that graph coarsening preserves data utility while enhancing privacy, making it preferable to traditional DP methods. Integrating HE metrics underscores the effectiveness of graph coarsening in balancing privacy and performance in FL.

## 4.3 EVALUATING UTILITY OF CPFL FOR PRIVACY ENHANCEMENT

The two subfigures, 4 (a) and 4 (b), illustrate how the percentage of nodes and edges concealed varies with increasing coarsening ratios ($r$) across three datasets: MUTAG, PROTEINS, and Reddit (M). As the coarsening ratio increases, a higher percentage of nodes and edges are concealed, indicating a reduced level of granularity in the graph representation. Concealing a greater proportion of nodes and edges by increasing the coarsening ratio directly reduces the level of detail in the graph. This results in a decrease in $L_2$-sensitivity, as each node or edge has a diminished influence on the graph's overall structure. Lower $L_2$-sensitivity means that the impact of modifying or omitting a single node or edge is minimized, making it harder for an adversary to distinguish whether a particular node or edge was part of the original graph. In differential privacy, the parameter $\epsilon$ quantifies the level of privacy provided. Lower $L_2$-sensitivity translates to a smaller $\Delta f$ (the sensitivity parameter), which means that for a fixed amount of noise, the effective $\epsilon$ value will decrease. As the figures show that a higher coarsening ratio results in more concealed nodes and edges, it implies a reduced sensitivity, thereby strengthening privacy guarantees (lower $\epsilon$).

$L_2$-sensitivity measures the maximum change in a function's output when a single input data point is modified. For graph data, it indicates how much model output or gradient values change when a specific node or edge is altered. By coarsening the graph, the number of nodes and edges is reduced, effectively lowering $L_2$-sensitivity, which decreases the potential for inferring specific details from the original graph structure. In the context of differential privacy, lower $L_2$-sensitivity means that achieving a given privacy level ($\epsilon$) requires less noise, thus preserving model utility. As observed in Figure 4(c), $L_2$-sensitivity decreases consistently with higher coarsening ratios for various datasets, confirming that graph coarsening can act as a natural privacy mechanism. This

Table 1: Classification performance on the single-dataset multi-client setting for $r = 0.5$ and $\epsilon=5$.

| | PROTIENS | | | IMDB (B) | | | DHFR | | | COLLAB | | |
|---|---|---|---|---|---|---|---|---|---|---|---|---|
| | All | CPFL | DP-SGD | All | CPFL | DP-SGD | All | CPFL | DP-SGD | All | CPFL | DP-SGD |
| Self-Train | 0.69 | 0.62 | 0.35 | 0.78 | 0.78 | 0.46 | 0.61 | 0.61 | 0.47 | 0.71 | 0.71 | 0.35 |
| FedAvg | 0.74 | 0.62 | 0.44 | 0.78 | 0.72 | 0.47 | 0.66 | 0.66 | 0.56 | 0.73 | 0.71 | 0.33 |
| FedProx | 0.75 | 0.62 | 0.62 | 0.77 | 0.72 | 0.63 | 0.75 | 0.65 | 0.61 | 0.69 | 0.68 | 0.66 |
| GCFL | 0.76 | 0.62 | 0.32 | 0.81 | 0.75 | 0.46 | 0.67 | 0.68 | 0.56 | 0.74 | 0.73 | 0.41 |
| GCFL+ | 0.75 | 0.62 | 0.35 | 0.77 | 0.74 | 0.47 | 0.71 | 0.68 | 0.53 | 0.73 | 0.73 | 0.38 |

Table 2: Classification performance on the multi-data multi-client setting

| **Molecules** | | | | | | | | | |
|---|---|---|---|---|---|---|---|---|---|
| | ALL | CPFL | | | | DP-SDG | | | |
| | | r=0.1 | r=0.2 | r=0.3 | r=0.5 | $\epsilon$=8 | $\epsilon$=5 | $\epsilon$=4 | $\epsilon$=3 |
| Self-Train | 0.7 | 0.63 | 0.63 | 0.64 | 0.67 | 0.62 | 0.58 | 0.58 | 0.56 |
| FedAvg | 0.72 | 0.63 | 0.63 | 0.63 | 0.66 | 0.62 | 0.62 | 0.62 | 0.62 |
| FedProx | 0.73 | 0.63 | 0.62 | 0.55 | 0.6 | 0.67 | 0.67 | 0.67 | 0.67 |
| GCFL | 0.72 | 0.63 | 0.63 | 0.63 | 0.67 | 0.67 | 0.49 | 0.5 | 0.5 |
| GCFL+ | 0.72 | 0.63 | 0.63 | 0.63 | 0.68 | 0.62 | 0.62 | 0.62 | 0.62 |
| **Proteins** | | | | | | | | | |
| | ALL | CPFL | | | | DP-SDG | | | |
| | | r=0.1 | r=0.2 | r=0.3 | r=0.5 | $\epsilon$=8 | $\epsilon$=5 | $\epsilon$=4 | $\epsilon$=3 |
| Self-Train | 0.58 | 0.5 | 0.5 | 0.5 | 0.52 | 0.44 | 0.44 | 0.44 | 0.44 |
| FedAvg | 0.55 | 0.44 | 0.51 | 0.52 | 0.54 | 0.43 | 0.43 | 0.36 | 0.36 |
| FedProx | 0.58 | 0.49 | 0.51 | 0.47 | 0.51 | 0.5 | 0.47 | 0.46 | 0.45 |
| GCFL | 0.57 | 0.45 | 0.52 | 0.53 | 0.52 | 0.43 | 0.43 | 0.44 | 0.44 |
| GCFL+ | 0.57 | 0.44 | 0.51 | 0.51 | 0.54 | 0.43 | 0.36 | 0.36 | 0.38 |
| **Social Networks** | | | | | | | | | |
| | ALL | CPFL | | | | DP-SDG | | | |
| | | r=0.1 | r=0.2 | r=0.3 | r=0.5 | $\epsilon$=8 | $\epsilon$=5 | $\epsilon$=4 | $\epsilon$=3 |
| Self-Train | 0.64 | 0.6 | 0.62 | 0.62 | 0.62 | 0.45 | 0.45 | 0.45 | 0.45 |
| FedAvg | 0.66 | 0.6 | 0.61 | 0.6 | 0.61 | 0.48 | 0.46 | 0.46 | 0.45 |
| FedProx | 0.66 | 0.62 | 0.63 | 0.64 | 0.63 | 0.62 | 0.62 | 0.59 | 0.57 |
| GCFL | 0.65 | 0.61 | 0.61 | 0.61 | 0.61 | 0.51 | 0.47 | 0.45 | 0.46 |
| GCFL+ | 0.65 | 0.6 | 0.61 | 0.62 | 0.62 | 0.49 | 0.38 | 0.38 | 0.36 |

reduction in $L_2$-sensitivity provides strong empirical support for our claim that graph coarsening is an effective privacy-preserving measure in federated learning. By lowering sensitivity, graph coarsening minimizes the risk of information leakage even if an attacker has access to the coarsened graph or the shared model updates. In practical federated learning scenarios, adjusting the coarsening ratio allows for fine-tuning the balance between privacy and accuracy. Higher coarsening ratios offer stronger privacy protection while maintaining acceptable performance levels.

Thus, graph coarsening not only aligns with the principles of $\epsilon$-differential privacy but also provides a straightforward, efficient approach to achieve tighter privacy guarantees with minimal computational overhead. This makes it a promising technique for enhancing privacy in real-world federated learning applications.

Table 3: Classification performance on the multi-domain multi-data multi-client setting

| **Mix 1 (Molecules + Proteins)** | | | | | | | | | |
|---|---|---|---|---|---|---|---|---|---|
| | ALL | CPFL | | | | DP-SDG | | | |
| | | r=0.1 | r=0.2 | r=0.3 | r=0.5 | $\epsilon$=8 | $\epsilon$=5 | $\epsilon$=4 | $\epsilon$=3 |
| Self-Train | 0.66 | 0.58 | 0.58 | 0.58 | 0.62 | 0.54 | 0.54 | 0.53 | 0.53 |
| FedAvg | 0.65 | 0.56 | 0.58 | 0.59 | 0.62 | 0.58 | 0.55 | 0.53 | 0.53 |
| FedProx | 0.69 | 0.56 | 0.59 | 0.59 | 0.61 | 0.61 | 0.6 | 0.6 | 0.59 |
| GCFL | 0.64 | 0.56 | 0.58 | 0.59 | 0.63 | 0.58 | 0.58 | 0.55 | 0.54 |
| GCFL+ | 0.67 | 0.56 | 0.58 | 0.59 | 0.62 | 0.49 | 0.49 | 0.48 | 0.48 |
| **Mix 2 (Molecules + Social Networks)** | | | | | | | | | |
| | ALL | CPFL | | | | DP-SDG | | | |
| | | r=0.1 | r=0.2 | r=0.3 | r=0.5 | $\epsilon$=8 | $\epsilon$=5 | $\epsilon$=4 | $\epsilon$=3 |
| Self-Train | 0.64 | 0.62 | 0.62 | 0.62 | 0.6 | 0.5 | 0.48 | 0.48 | 0.48 |
| FedAvg | 0.66 | 0.62 | 0.62 | 0.58 | 0.6 | 0.57 | 0.56 | 0.55 | 0.54 |
| FedProx | 0.66 | 0.63 | 0.65 | 0.59 | 0.65 | 0.65 | 0.65 | 0.64 | 0.63 |
| GCFL | 0.65 | 0.63 | 0.64 | 0.56 | 0.6 | 0.58 | 0.58 | 0.58 | 0.53 |
| GCFL+ | 0.65 | 0.63 | 0.63 | 0.55 | 0.59 | 0.59 | 0.58 | 0.58 | 0.58 |

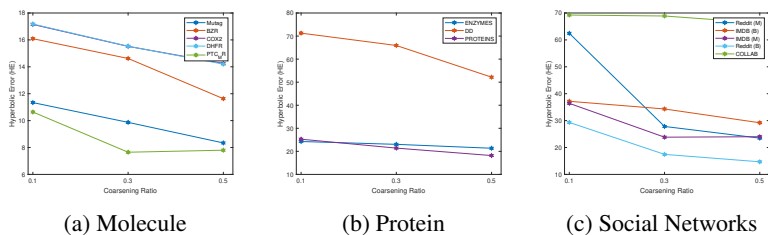

(a) Molecule          (b) Protein          (c) Social Networks

Figure 3: Hyperbolic error (HE) values from FGC across different coarsening ratios (r).

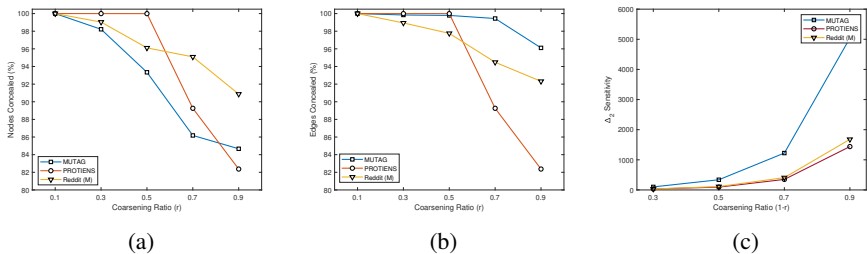

(a)          (b)          (c)

Figure 4: Evaluating graph coarsening for privacy enhancement in federated learning ensuring (a) node level privacy, (b) edge level privacy and (c) $L_2$ sensitivity for coarsening ratio (r).

## 4.4 CONVERGENCE ANALYSIS

We evaluated the classification accuracy relative to communication round to demonstrate convergence of the FL setup with the privacy measures implemented through CPFL and DP-SDG [29]. Figure 5 illustrates the average accuracy curve for datasets for different domains. It can be inferred that CPFL achieves a faster convergence rate compared to the widely used DP.

## 5 CONCLUSION

In this work, we introduced a graph coarsening technique as a privacy measure in federated learning environments for graph data. This approach facilitates private and secure collaborative training of advanced graph models, such as neural networks for classification, without the need for direct data exchange. Our extensive experiments show that the tradeoff associated with this privacy measure is minimal compared to traditional methods like differential privacy. Moreover, since this measure is applied as a preprocessing step, it incurs minimal computational overhead, ensuring that the performance of round-wise communication in federated learning remains unaffected. While our study primarily focuses on protecting against graph reconstruction attacks, future research should investigate other potential attacks in federated learning for graph data. We believe this work lays the groundwork for further studies on privacy-preserving techniques through client-side data manipulation and the development of methods to evaluate their effectiveness.

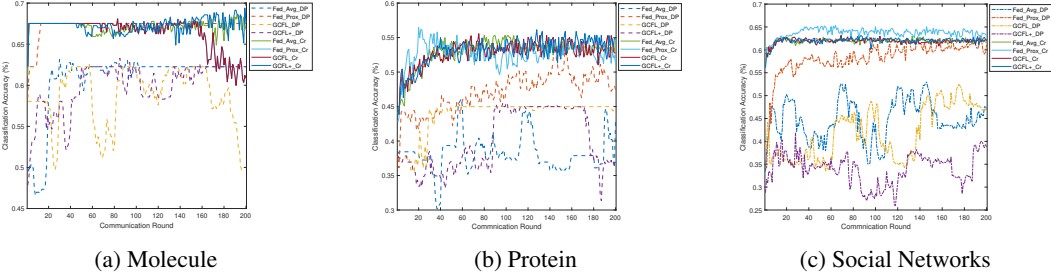

(a) Molecule          (b) Protein          (c) Social Networks

Figure 5: Classification accuracy with graph coarsening (r=0.5) and differential privacy ($\epsilon$=5) versus communication round in multi dataset multi=client setting.

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

## A  Appendix

### A.1  Dataset Description

We have utilized a diverse set of datasets from three distinct subcategories: Molecules, Proteins, and Social Networks. Each dataset varies in terms of the number of classes, graphs, average nodes, and average edges, providing a broad spectrum for analysis and classification tasks.

In the Molecules subcategory, we have the following datasets: Mutag, BZR, COX2, DHFR, and PTC_MR. Each of these datasets contains 2 classes. Mutag consists of 188 graphs with an average of 17.93 nodes and 19.79 edges per graph. BZR has 405 graphs, averaging 35.75 nodes and 38.36 edges per graph. COX2 includes 467 graphs with an average of 41.22 nodes and 43.45 edges per graph. DHFR comprises 756 graphs, with averages of 42.43 nodes and 44.54 edges per graph. PTC_MR has 344 graphs, with an average of 14.29 nodes and 14.69 edges per graph.

For the Proteins subcategory, we employed the ENZYMES, DD, and PROTEINS datasets. ENZYMES is the most diverse with 6 classes, encompassing 600 graphs, each averaging 32.63 nodes and 62.14 edges. DD has 2 classes and includes 1178 graphs with a high average of 284.32 nodes and 715.66 edges per graph. The PROTEINS dataset also has 2 classes, consisting of 1113 graphs, with averages of 39.06 nodes and 72.82 edges per graph.

In the Social Networks subcategory, we used datasets such as COLLAB, Reddit (M), IMDB (B), IMDB (M), and Reddit (B). COLLAB has 3 classes and includes 5000 graphs with an average of 74.49 nodes and 2457.78 edges per graph. Reddit (M) is more diverse with 5 classes, consisting of 5000 graphs with an average of 508.29 nodes and 594.87 edges per graph. IMDB (B) includes 2 classes and 1000 graphs, each averaging 19.77 nodes and 96.53 edges. IMDB (M) has 3 classes, encompassing 1500 graphs with averages of 13 nodes and 65.94 edges. Lastly, Reddit (B) contains 2 classes and 2000 graphs, each averaging 429.63 nodes and 497.75 edges.

| MOLECULES | | | | |
|---|---|---|---|---|
| | Number of Classes | Number of Graphs | Average Nodes | Average Edges |
| Mutag | 2 | 188 | 17.93 | 19.79 |
| BZR | 2 | 405 | 35.75 | 38.36 |
| COX2 | 2 | 467 | 41.22 | 43.45 |
| DHFR | 2 | 756 | 42.43 | 44.54 |
| PTC_MR | 2 | 344 | 14.29 | 14.69 |
| PROTEINS | | | | |
| | Number of Classes | Number of Graphs | Average Nodes | Average Edges |
| ENZYMES | 6 | 600 | 32.63 | 62.14 |
| DD | 2 | 1178 | 284.32 | 715.66 |
| PROTEINS | 2 | 1113 | 39.06 | 72.82 |
| SOCIAL NETWORKS | | | | |
| | Number of Classes | Number of Graphs | Average Nodes | Average Edges |
| COLLAB | 3 | 5000 | 74.49 | 2457.78 |
| Reddit (M) | 5 | 5000 | 508.29 | 594.87 |
| IMDB (B) | 2 | 1000 | 19.77 | 96.53 |
| IMDB (M) | 3 | 1500 | 13 | 65.94 |
| Reddit (B) | 2 | 2000 | 429.63 | 497.75 |

### A.2  Settings

**Single Data Multi-Client (SDMC) Setting** In the SDMC setting, we have a single graph dataset distributed across multiple clients. Each client holds a portion of the graph data. The goal is to train a graph machine learning model while respecting data privacy and without centralizing the data.

**Multi Data Multi-Client (MDMC) Setting** In the MDMC setting, we have multiple graph datasets, each owned by different clients. The goal is to train a federated graph machine learning model that can generalize across different graphs while maintaining data privacy.

