# OpenReview forum: "Coarsening to Conceal: Enabling Privacy-Preserving Federated Learning for Graph Data"
_ICLR.cc/2025/Conference — Submitted to ICLR 2025_

### Official Review · Reviewer_xxzj · 2024-10-23

**Soundness:** 1
**Presentation:** 2
**Contribution:** 2
**Rating:** 3
**Confidence:** 4

**Summary:**

This paper proposes CPFL, a federated learning (FL) framework integrated with the graph coarsening technique for privacy-preserving federated graph classification. CPFL leverages featured graph coarsening (FGC) to compress graphs locally at each client, followed by conventional FL workflow. In experiments, CPFL can consistently outperform DP-SGD in different settings.

**Strengths:**

The reviewer appreciates the simplicity of the proposed methodology. It seems straightforward and effective.

**Weaknesses:**

1. The novelty of this paper is quite limited. CPFL is basically a naïve combination of FGC and other FL frameworks.
2. The technical quality of this paper is not good enough. Without proposing brand new methodologies, the paper provides limited theoretical contributions, lacks baseline comparisons, and has some unexplained self-contradictory or counter-intuitive claims/conclusions.
3. The clarity of this paper can be further improved. There are many lengthy and repetitive sentences/paragraphs that can be removed to save space for more important content.

**Questions:**

1. The graph coarsening method in CPFL seems to be just FGC. Is there any innovation in CPFL except the combination of graph coarsening and federated learning?
2. In conventional FL frameworks, clients would only share model weights/updates to the server. Why do the authors assume that the attacker (the honest-but-curious server) has access to the coarsened graph dataset?
3. Is there any proof that provides theoretical guarantees on the difficulties of reconstructing the original graphs from its coarsened version?
4. The analysis in Section 3.4 is not rigorous enough and may need to be reformulated. After reading through this section, the reviewer still does not know whether it is good or bad to have higher sensitivity in terms of privacy preservation.
5. Based on the experimental results and the authors' claim, higher coarsening ratios offer stronger privacy protection and better model performance. How is this possible?
6. It is better to report the costs (running time, RAM usage, etc.) of the graph coarsening step.
7. There is only one baseline method (DP-SGD). The authors may consider adding more, such as sparsification and condensation techniques.
8. CPFL currently only supports graph-level tasks. The authors may need to think about how to extend it to support node/edge-level tasks in the future.

---

### Official Review · Reviewer_aFqN · 2024-10-30

**Soundness:** 1
**Presentation:** 2
**Contribution:** 2
**Rating:** 3
**Confidence:** 4

**Summary:**

This paper presents an approach named CPFL designed to preserve privacy while minimizing model performance degradation in FedGNN (Federated Graph Neural Networks. The authors leverage the existing graph coarsening method FGC, adapting it for the FedGNN context. Theoretical analysis suggests that this coarsening technique effectively defends against malicious attempts to reconstruct private data by adversaries. Empirical results indicate that CPFL outperforms traditional differential privacy methods, specifically DP-SGD, in terms of maintaining model utility. However, the paper primarily focuses on applying FGC to FedGNN, with baseline comparisons limited to DP-based methods.

**Strengths:**

1. The theoretical analysis of CPFL provides a theoretical explanation that the adversary cannot recover the private data from the coarsened graph feature.
2. Experimental results show that CPFL maintains higher model utility compared to DP-SGD.

**Weaknesses:**

1. The core algorithm primarily involves applying the existing FGC method within the FedGNN framework, with subsequent steps adhering closely to the standard FedAvg approach. This application, while effective, may not introduce substantial methodological innovations beyond the integration of FGC.
2. The threat model is not practical. The attacker should be one of the participants, the third party, or the server. Thus, it is not practical for any of them to access the coarsened data without accessing the original data. A practical attacker may access the model parameters.
3. The statement "Thus we can say that the two dataset $G(V, E)$ and $G_r(V_r, E_r)$ are differentially private" appears to be inaccurate. The coarsened graph and the original graph are not exact neighbors, and there is no definitive guarantee that the model's output satisfies differential privacy under this transformation. Clarification or revision of this theoretical claim is necessary.
4. Certain aspects of the theoretical analysis are unclear. For example, the coarsening ratio $r$ is initially defined as m/p but later referred to as k/P.
5. The experimental evaluation compares CPFL solely with DP-SGD. Including additional privacy-preserving methods such as Secure Multi-Party Computation, Homomorphic Encryption, or noise obfuscation techniques would provide a more comprehensive assessment of CPFL's performance and its computational and communication overheads relative to other approaches.
6. The paper does not clearly delineate the relationship between CPFL's parameters and traditional differential privacy parameters. For instance, as the privacy parameter $\epsilon$ increases, model utility degradation decreases, but this may make it easier for attackers to reconstruct data. Including analysis or conducting privacy attack experiments to explore this relationship would strengthen the study.

**Questions:**

The theoretical analysis suggests that the coarsening process is irreversible, leading to information loss that adversely affects model utility. Could the authors quantify the extent of information loss? Is there a theoretical or empirical bound on this information loss, and how does it impact the overall model performance?

---

### Official Review · Reviewer_qatG · 2024-11-01

**Soundness:** 2
**Presentation:** 2
**Contribution:** 2
**Rating:** 3
**Confidence:** 4

**Summary:**

This paper presents a graph coarsening method for a privacy-preserving federated learning framework (CPFL), aimed at enhancing the privacy of federated learning (FL) on graph data. This method employs graph coarsening techniques, which simplify the graph structure to mitigate the risk of sensitive information leakage. By aggregating nodes and edges into supernodes and superedges, it becomes difficult to reconstruct the original data, while ensuring that the coarsening method satisfies differential privacy. Extensive experiments on multiple datasets demonstrate that CPFL maintains model performance while preserving privacy.

**Strengths:**

1. This paper applies graph coarsening to the privacy-preserving task of federated learning, providing a novel perspective for traditional FL privacy methods.
2. The experiments utilized various datasets from molecules, proteins, and social networks to demonstrate the broad applicability and effectiveness of CPFL.
3. The experimental results indicate that CPFL can maintain model performance to a certain extent while providing privacy protection, and in some cases, it even outperforms differential privacy methods.

**Weaknesses:**

1. The discussion on the technical innovation of the method is insufficient. For the framework, graph coarsening is applied only to the client. This raises the question: Is there a bottleneck in transferring the graph coarsening technique from centralized to distributed scenarios? Is this the first work to apply graph coarsening in FL? I also have another hypothesis, that this study aims to reveal the privacy-preserving capability of graph coarsening. Please elaborate on the differences in privacy protection that the proposed method achieves compared to existing graph coarsening techniques.
2. The statement that coarsening can satisfy differential privacy is not rigorous, lacking theoretical and experimental analysis. The paper mostly verifies the privacy protection effect of graph coarsening through experiments. However, for the attack cases proposed in the paper, there is no experimental proof that CPFL can resist these attacks while maintaining performance.
3. The experimental design and result analysis are insufficient. For example, in Table 1, why does FedProx show the same performance in CPFL and DP-SGD (both 0.62)? Why does GCFL+ show the same performance in "All" and CPFL (both 0.73)? In Table 2, does r=0.1 in CPFL correspond to the results of DP-SGD with $\epsilon$=8? Further parameter analysis is needed to clarify, even if DP methods generally perform much worse than the proposed method. CPFL only loses a small amount of performance, how can it be proven that the protection capability of graph coarsening is not limited? Please further provide relevant attack experiments mentioned in point 2 to demonstrate the actual privacy protection.

**Questions:**

1. What optimizations or improvements has CPFL made to the graph coarsening technique?
2. Can the privacy-preserving capability of CPFL surpass that of DP?
3. What does CPFL sacrifice to achieve privacy protection?

---

### Official Review · Reviewer_jkbG · 2024-11-06

**Soundness:** 2
**Presentation:** 2
**Contribution:** 2
**Rating:** 3
**Confidence:** 5

**Summary:**

This paper studies the problem of federated graph learning. Specifically, the authors propose to use graph coarsening methods like FGC to aggregate graph information, in order to satisfy the privacy requirement of federated learning. A discussion about graph coarsening and DP is presented in the paper, and simulation results are provided to show the performance of the proposed method.

**Strengths:**

The idea of using graph coarsening might be an effective way to help improve the performance of federated graph learning.

**Weaknesses:**

There are three major issues with this paper:

1. After reading the whole paper, I do not see why graph coarsening is needed in the learning framework that the authors proposed. In Figure 2, it seems that the only information that the server uses is the model parameters uploaded from each client (i.e., the server does model average like FedAvg), then why do we bother sending the coarse graphs $\tilde{\mathcal{D}_k}$ in Equation (7)? The white-box assumption "We assume a robust malicious entity, such as an honest-but-curious server, which lacks direct access to G but has white-box access to both the coarsened graph dataset and the model" does not hold if we do not send $\tilde{\mathcal{D}_k}$ to the server. I do not see the necessity of graph coarsening here, as we can just aggregate the model parameters without privacy concerns. If in practice one sees graph coarsening + FedAvg performs better than plain vanilla FedAvg on some datasets, it would be because graph coarsening serves as a graph denoising step that improves the model performance on each client anyhow, which has nothing to do with FL. In general, I do not believe training on graph coarsening + FedAvg would outperform FedAvg, as the coarse graph is not used at all in the proposed FL framework. Nevertheless, I do think graph coarsening could be helpful if some new designs are adopted on the server side, but not in the current framework.

2. The theory part of this paper is very weak. The proof of the privacy guarantees is described in words instead of rigorous derivations. For example, line 303-305 "This signifies that the reconstructed graph retains a similar number of nodes as the original graph, but contains more edges compared to the original graph. Thus we can say that the two dataset G(V, E) and Gr(Vr, Er) are differentially private.", this is not how edge-DP is proved, and if it is really the case, what is the value of $\varepsilon$ in this case? Furthermore, I do not quite understand what message line 308-360 wants to convey. The sensitivity analysis is used in the wrong way and the conclusion is confusing.

3. The whole framework is a simple combination of FGC + FedAvg applied to federated graph classification problems without in-depth explanations, making the contribution of this paper limited.

One minor comment is that the notations in the paper should be consistent. For example, on page 7, I believe $P$ should be $p$, $d$ should be $n$, and $k$ should be $m$.

**Questions:**

How is the proposed method different from training local models based on FGC local graphs + FedAvg on the server side?

---

### Meta-Review · Area_Chair_kUuD · 2024-12-07

**Metareview:**

The paper introduces CPFL, a federated learning framework incorporating graph coarsening for privacy-preserving federated graph classification. The key claim is that applying Featured Graph Coarsening locally on client data enhances privacy by reducing the risk of sensitive information leakage. The authors assert that this approach maintains competitive performance compared to conventional FL methods while offering superior privacy protection relative to differential privacy mechanisms like DP-SGD. Experimental results on datasets from molecular, protein, and social network domains are presented.

**The Strengths of the Paper**

Novel Contextual Application: The use of graph coarsening in a federated learning setting introduces an interesting perspective on privacy preservation in distributed graph-based systems.

Performance Demonstrations: The experimental results show that CPFL can maintain model performance and, in some cases, outperform DP-SGD in terms of utility.

Cross-Domain Evaluation: Multiple datasets from distinct domains enhance the generalizability of the empirical claims.

**Weaknesses and Missing Elements**

Limited Technical Novelty: CPFL appears to be a straightforward integration of existing graph coarsening methods with standard FL frameworks like FedAvg.

Weak Theoretical Foundation: The theoretical claims, particularly around privacy guarantees, are weak.

Inadequate Clarity and Analysis: Several experimental results are counterintuitive or unexplained, such as the consistent performance of FedProx across setups and the relationship between coarsening ratio and utility/privacy. Additionally, practical details like computational overhead of coarsening are missing.

Given the unanimous agreement among reviewers to reject the paper, I concur with this assessment.

**Additional Comments On Reviewer Discussion:**

N/A

---

### Decision · Program_Chairs · 2025-01-22

Reject